# Integrated Transcriptomic and Proteomic Characterization of a Chromosome Segment Substitution Line Reveals the Regulatory Mechanism Controlling the Seed Weight in Soybean

**DOI:** 10.3390/plants13060908

**Published:** 2024-03-21

**Authors:** Siming Wei, Zhenhai Yu, Fangfang Du, Fubin Cao, Mingliang Yang, Chunyan Liu, Zhaoming Qi, Qingshan Chen, Jianan Zou, Jinhui Wang

**Affiliations:** 1National Key Laboratory of Smart Farm Technology and System, Key Laboratory of Soybean Biology in Chinese Ministry of Education, College of Agriculture, Northeast Agricultural University, Harbin 150030, China; wsm20016@126.com (S.W.); s220301027@neau.edu.cn (F.D.); fbincao@126.com (F.C.); mlyang@neau.edu.cn (M.Y.); cyliucn@neau.edu.cn (C.L.); qizhaoming1860@126.com (Z.Q.);; 2Heilongjiang Province Green Food Science Institute, Harbin 150028, China; hyzh1188@sina.com

**Keywords:** seed weight, seed size, regulatory pathway, transcriptomic, proteomic, soybean

## Abstract

Soybean is the major global source of edible oils and vegetable proteins. Seed size and weight are crucial traits determining the soybean yield. Understanding the molecular regulatory mechanism underlying the seed weight and size is helpful for improving soybean genetic breeding. The molecular regulatory pathways controlling the seed weight and size were investigated in this study. The 100-seed weight, seed length, seed width, and seed weight per plant of a chromosome segment substitution line (CSSL) R217 increased compared with those of its recurrent parent ‘Suinong14’ (SN14). Transcriptomic and proteomic analyses of R217 and SN14 were performed at the seed developmental stages S15 and S20. In total, 2643 differentially expressed genes (DEGs) and 208 differentially accumulated proteins (DAPs) were detected at S15, and 1943 DEGs and 1248 DAPs were detected at S20. Furthermore, integrated transcriptomic and proteomic analyses revealed that mitogen-activated protein kinase signaling and cell wall biosynthesis and modification were potential pathways associated with seed weight and size control. Finally, 59 candidate genes that might control seed weight and size were identified. Among them, 25 genes were located on the substituted segments of R217. Two critical pathways controlling seed weight were uncovered in our work. These findings provided new insights into the seed weight-related regulatory network in soybean.

## 1. Introduction

Globally, soybean is widely used as a crucial food, oil, and forage crop. In 2023, the planting area of soybeans in China was 157 million mu, and the total production was 20.84 million tons, with a self-sufficiency rate of only 17.3% (http://www.stats.gov.cn/, accessed on 12 March 2024). Improving soybean yield has always been a critical goal worth continuously pursuing. Seed yield is determined by the number of seeds produced and seed weight and is a complex quantitative trait [1,2]. The 100-seed weight, seed length, and seed width are critical factors determining soybean yield. Seed-related traits are often correlated with each other. In previous studies, the 100-seed weight and seed size were generally positively correlated [3,4,5,6,7]. Cell development determines the final seed size and weight, and cell proliferation and expansion co-regulate the final seed size [2]. Genes regulating seed weight and size were recently cloned in soybean [2,3,4,5,6,7,8,9]. Revealing the seed weight regulation pathway contributes to the genetic improvement in soybean.

Transcriptomic analysis is an effective method for understanding the molecular genetic relationship of seed weight based on the transcriptional variation. The genes related to seed weight and size control *GmGA20OX*, *SoyWRKY15a*, and *GmCYP78A5* were cloned based on the results of the transcriptomic analysis [3,10,11]. The potential genetic variation underlying the developmental evolution of soybean seeds was revealed based on the transcriptomic comparison between the cultivar and wild soybean [12]. Proteomic analysis is generally used for investigating the effects of seed development on the yield, thereby predicting potential proteins related to seed weight and size and revealing the interaction network between seed size-related proteins in major crops [13,14,15,16,17]. Multi-omics are more efficient and comprehensive for revealing the regulatory mechanism underlying seed development. Integrated transcriptomic and proteomic analyses were recently used to investigate the potential pathways related to seed weight and size [11,18,19].

The molecular networks of seed size have been preliminarily established in *Arabidopsis* and rice [20,21]. Mitogen-activated protein kinase (MAPK) signaling pathways are the critical seed size-regulating pathways [21]. The relatively complete MAPK cascade, OsMKKK10-OsMKK4-OsMPK6 module, has been reported to control grain size in rice, and this module was identified through the genetic screening of the small grain phenotype and map-based cloning [22,23,24]. The loss of function of *OsMKKK10*, *OsMKK4*, or *OsMPK* results in small and light grains, and *OsMKKK10* or *OsMKK4* expression results in heavy and large grains [25]. Similarly, in *Arabidopsis*, *MKK4*, and *MKK5* control seed size upstream of *MPK6*, which is consistent with the regulatory pattern in rice [26]. *ERECTA1* upstream of the OsMKKK10-OsMKK4-OsMPK6 cascade is involved in controlling seed size by regulating cytokinin homeostasis in rice [27]. Moreover, the ERECTA-MAPK-DA1-UBP15 signaling module regulates seed size in *Arabidopsis* [28]. A lectin RLK, LecRK-VIII.2, controls the seed size, silique number, seed number, and seed yield in *Arabidopsis*, and the positive role of LecRK-VIII.2 in seed coat is a maternal effect exerted through the simultaneous control of cell expansion and proliferation [29]. The OsMKK-OsMAPK6-OsWRKY53 signaling module was identified to control seed size in rice through cell expansion [30,31]. In addition, Map Kinase Phosphatase 1 (OsMKP1) was involved in seed size control by negatively regulating the OsMKKK10-OsMKK4-OsMPK6 cascade [27]. These reports suggest that the OsMKKK10-OsMKK4-OsMPK6 module for seed size control is conserved among multiple species.

Cell walls are essential for plant organ development [32]. Plant cell walls are basically composed of three major polysaccharide types, including pectin, cellulose, and hemicellulose [33]. The cell wall structure and composition determine its various functions, and multiple functions have been identified for cell wall biosynthesis and modification pathways [34]. One of the crucial functions of cell walls is to participate in seed development, and ultimately determine seed size [35,36,37,38,39]. Cell wall biosynthesis and modification-related genes that control seed weight have been identified in rice, *Arabidopsis*, maize, barley, wheat, tomato, and soybean [5,35,36,37,38,39,40,41,42,43,44,45,46]. Few studies have reported that structural variations in the cell wall affect seed development, and ultimately determine seed size in soybean. The seed thickness-regulating gene *ST1* (*Seed Thickness 1*) encodes uridine diphosphate (UDP)-D-glucuronate 4-epimerase. *ST1* can catalyze UDP-galacturonic acid production and promote pectin biosynthesis [5]. In addition, the major function of polygalacturonase is to decompose pectin or pectinase enzymes. *PG031* is a pleiotropic *polygalacturonase* (*PG*) gene that regulates seed coat permeability and 100-seed weight in soybean. Subcellular localization has demonstrated that PG031 is localized to the cell wall of the seed coat [46]. These studies have shown that cell wall-related pathways are crucial for controlling seed size. Therefore, as more seed size-regulating genes are cloned, the molecular genetic network of the seed size needs to be revealed.

CSSLs are ideal genetic resources for studying genetic variations because each CSSL carries a single chromosomal segment. CSSLs are used to identify the quantitative trait loci (QTL), cloning genes, identifying the genetic pattern of agronomic traits, and investigating the interacting relationships between genes and the environment. CSSLs were first developed from tomato plants [47] and were later widely used to study multiple crops such as rice, barley, and cotton [48,49,50]. A CSSL population was developed by substituting the chromosome segments of ‘ZYD00006’ into the SN14 genome [51,52]. A large-seed CSSL R217 was selected from among the CSSLs. In this study, the regulatory pathways of seed weight were illuminated based on the results of the integrated transcriptomic and proteomic analyses.

## 2. Results

### 2.1. Phenotypic Variation of the Large Seed CSSL R217

A large-seed CSSL R217 was selected from a wild soybean CSSLs population. The CSSL was developed from the wild soybean varieties ‘ZYD00006’ (ZYD06) and ‘Suinong14’ (SN14), which were used as the donor and recurrent parents, respectively. The substituted segments of CSSLs covered over 90% of the wild soybean genome [51,52]. Phenotypic statistics of seed weight and size for multiple years indicated that 100-seed weight, seed length, seed width, and seed weight per plant of R217 increased compared to SN14 (Figure 1A–C and Appendix A). Therefore, R217 is a suitable material for studying the genetic variation mechanism of seed weight. These results suggest that chromosomal segments on R217 are the main reason for the variation in seed weight between R217 and SN14.

### 2.2. Transcriptomic Comparison between R217 and SN14

To reveal the transcriptional variations between R217 and SN14, the transcriptomic analysis was performed at S15 and S20 seed developmental stages. A total of 2643 DEGs were detected, including 1216 upregulated and 1427 downregulated DEGs between R217 and SN14 at S15 (Figure 2A). A total of 1934 DEGs were estimated at S20, including 251 upregulated and 1083 downregulated DEGs, respectively (Figure 2B). Overall, 2427 specific DEGs at S15, 1717 specific DEGs at S20, and 217 DEGs were shared between R217 and SN14 at both S15 and S20 (Figure 2C). The results of the KEGG enrichment analysis indicated that the downregulated DEGs at S15 were chiefly enriched in neutrophil extracellular trap formation, cell cycles, and phenylpropanoid biosynthesis (Figure 2D). Moreover, the upregulated DEGs at S15 were enriched in cytochrome P450, glycolysis/gluconeogenesis, Toll-like receptor signaling pathway (Figure 2E). At S20, the downregulated DEGs were enriched in the MAPK signaling pathway, RAS signaling pathway, and Toll-like receptor signaling pathway (Figure 2F), and the upregulated DEGs were enriched in the mRNA surveillance pathway, glycolysis/gluconeogenesis, zeatin biosynthesis (Figure 2G). The pathways, including Toll-like receptor signaling pathway, toll and IMD signaling pathway, starch and sucrose metabolism, MAPK signaling pathway-plant, starch and sucrose metabolism, NF-kappa B signaling pathway, and alpha-linolenic acid metabolism, were enriched in both upregulated and downregulated DEGs at S15 (Figure 2D,E). The processes of MAPK signaling pathway, phenylpropanoid biosynthesis, starch and sucrose metabolism, pyruvate metabolism, flavonoid biosynthesis, toll and IMD signaling pathway, Toll-like receptor signaling pathway, cytochrome P450, glycoly-sis/gluconeogenesis, NF-kappa B signaling pathway, amino sugar metabolism, nucleotide sugar metabolism, and RAS signaling pathway were together enriched at S15 and S20 (Figure 2D–G). Additionally, according to the GO enrichment analysis, histone H3-K9 modification and methylation, DNA modification, ion transport and methylation, plant-type secondary cell wall biogenesis, and cell cycle-related GO terms were enriched at S15 (Appendix A). Immune and plant-type cell wall modification-related GO terms were enriched at S20 (Appendix A). These results were worth mentioning that both GO and KEGG enrichment analyses were enriched in pathways and GO terms related to cell wall biosynthesis and modification, including phenylpropanoid biosynthesis, plant-type secondary cell wall biogenesis, and plant-type cell wall modification at two seed development stages (Figure 2 and Appendix A).

### 2.3. Proteomic Analysis of R217 and SN14

Proteomic analysis was performed to compare the variation in protein expression between R217 and SN14. In total, 65 and 143 downregulated and upregulated DAPs were detected at S15, respectively (Figure 3A). At S20, 803 and 445 downregulated and upregulated DAPs were detected, respectively (Figure 3B). Protein expression during soybean seed developmental stages is a dynamic process. Protein expression differed at S15 and S20 (Figure 3C,D). At S15, the response to stress-related GO terms was mainly enriched (Figure 3E). However, at S20, cell wall biosynthesis- and modification-related GO terms were mainly enriched, including pectin catabolic or metabolic process, cell wall modification, and cell wall organization or biogenesis at S20, as determined based on the GO enrichment analysis of DAPs (Figure 3F). Similarly, the cell wall-related pathways, including flavonoid biosynthesis and phenylpropanoid biosynthesis, were also enriched at both S15 and S20, as determined based on the KEGG enrichment analysis (Figure 3G,H); the MAPK signaling pathway was also enriched (Figure 3H). These results are consistent with the analysis of DEGs, which indicates that DAPs are also enriched in the MAPK signaling pathway and cell wall biosynthesis and modification. These results suggest that the variation in the seed weight in R217 is caused by differences in the expression patterns of genes related to the MAPK signaling pathway and cell wall biosynthesis and modification during seed developmental stages.

**Figure 2 plants-13-00908-f002:**
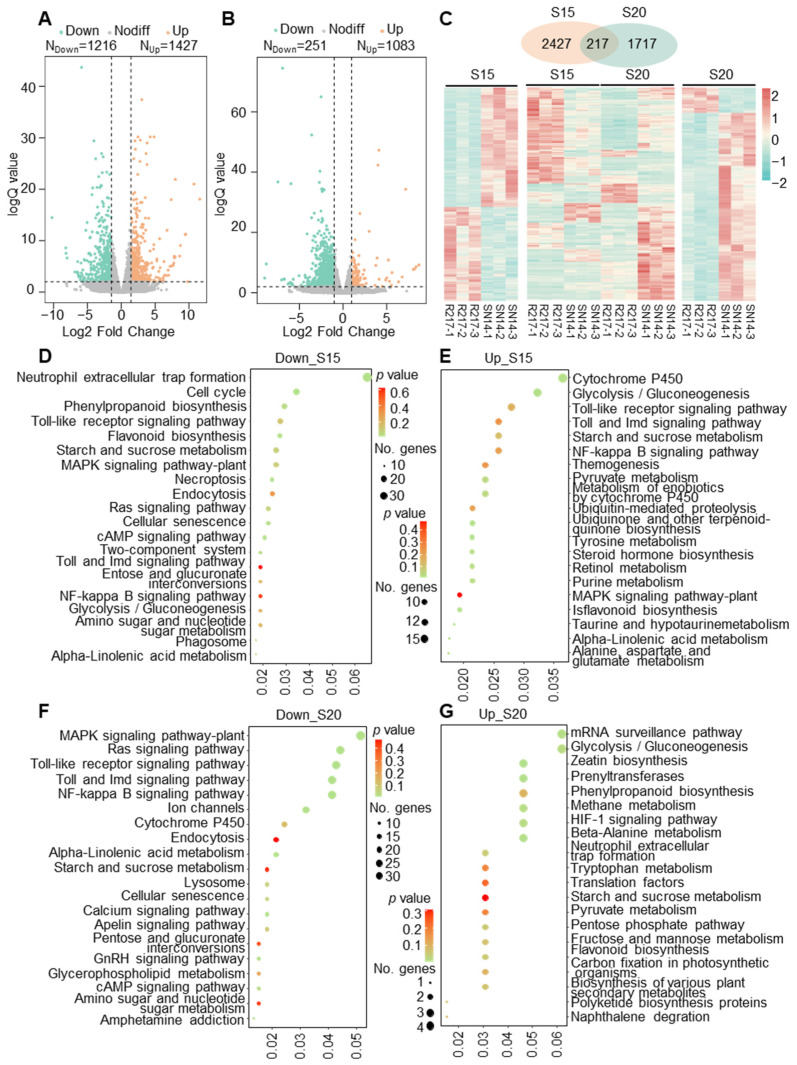
The DEG analysis and KEGG enrichment analysis in R217 and SN14 at S15 and S20 seed developmental stages. (**A**,**B**) Volcano map of DEGs at S15 and S20 seed developmental stages, respectively. Down—DEGs with downregulated expression. Up—DEGs with upregulated expression. Nodiff—genes without differential expression in RNA-seq analysis. (**C**) The heatmap of all DEGs in R217 and SN14 at S15 and S20 seed developmental stages. (**D**,**E**) The KEGG enrichment analysis of the top downregulated and upregulated DEGs at the S15 seed developmental stage. (**F**,**G**) The KEGG enrichment analysis of the top downregulated and upregulated DEGs at the S20 seed developmental stage. In (**D**–**G**), the vertical coordinates represent the GO terms enriched with genes from the substituted segments. The horizontal axis represents the gene ratio. No. genes—the number of genes.

### 2.4. Analysis of Genes on the Substituted Chromosomal Segments of R217

To further explain the seed weight variation of R217, the bin-map data of SN14 and ZYD00006 were used to obtain the information of substituted segments on genome of R217 [52]. Ten substituted segments were distributed on eight chromosomes of R217 (Figure 4A; Appendix A). The expression patterns of all genes on the substituted segments were analyzed based on the RNA-seq and proteomic data (Figure 4B–D). A total of 3203 genes were located on the substituted segments of R217 at S15 and S20 seed developmental stages (Figure 4B). These genes are mainly enriched in ion transport, glycometabolism and MAPK signaling pathways by performing GO enrichment analysis (Figure 4C). At the same time, the pathways of starch and sucrose metabolism, endocytosis, Toll-like receptor signaling pathway, and MAPK signaling pathway were mainly enriched based on KEGG enrichment analysis (Figure 4D). The enrichment analysis results of both methods were enriched in the MAPK signaling pathway.

### 2.5. Potential Molecular Genetic Mechanism of Seed Weight

Integrating transcriptomic and proteomic analyses provides a more comprehensive understanding of the molecular genetic mechanism of seed weight. The results of transcriptomic and proteomic analyses indicate that multiple pathways are involved in regulating seed weight. The variations in gene expression on the substituted segments of R217 caused changes in the expression patterns of DAPs and DEGs outside the substituted segments. Therefore, according to the results of gene analyses on the substituted segments, the MAPK signaling pathway might be a critical pathway regulating seed weight. Fourteen DEGs involved in the MAPK signaling pathway were selected from the substituted segments of R217 (Figure 5A). Furthermore, the analysis results of the DEGs and DAP expression patterns based on FPKM and LFQ intensity data indicate that 2 DAPs and 18 DEGs are involved in the MAPK signaling pathway in regions outside of the substituted segments (Figure 5A). The expression patterns of most MAPK signaling pathway-related genes were similar. The expression of these genes was significantly lower at S15 than at S20. The expression of the 18 DEGs decreased in R217 compared with SN14 (Figure 5A). Conversely, I1KVS8 and *Glyma.08G004500* were highly expressed at S15, but their expression was lower at S20 (Figure 5A). The expression level was higher in R217 than in SN14 at both S15 and S20 (Figure 5A).

The cell wall biosynthesis process might be another crucial pathway that participates in seed weight regulation. Two DAPs and nine DEGs involved in cell wall biosynthesis and modification were selected from the substituted segments of R217 (Figure 5B). The expression patterns of DEGs and DAPs were quite different between R217 and SN14 at S15 and S20. This indicates that DAPs and DEGs involved in the MAPK signaling pathway and cell wall biosynthesis and modification were direct factors affecting the variation in seed weight. Futhermore, the analysis results of gene expression patterns based on FPKM and LFQ intensity data indicate that nine DAPs and five DEGs outside the substituted segments were involved in cell wall biosynthesis and modification (Figure 5B). Among the aforementioned five DEGs, the expression patterns of four DEGs, namely *Glyma.06G225500*, *Glyma.09G051100*, *Glyma.15G157100*, and *Glyma.19G033600* were consistent, with the expression being higher at S15 than at S20 (Figure 5B). The expression level of all five DEGs significantly increased in R217 compared with SN14 at S15 and S20 (Figure 5B). The expression of I1JQY5, I1KAS1, K7LZ56, I1LVD2, and A0A0R0G1U8 was consistent with that of the aforementioned four DEGs (Figure 5B). On the contrary, the expression of four DAPs, namely I1N6E4, I1KDI6, I1L158, and I1L5A3, was lower at S15 than at S20, and the expression levels of the four proteins significantly decreased in R217 compared with SN14 (Figure 5B). The detailed expression and protein abundance data of these candidate genes are also listed (Appendix A). The results revealed that DEGs and DAPs related to the MAPK signaling pathway and the cell wall biosynthesis and modification pathway may be the main genes and proteins involved in seed weight regulation. Moreover, these results suggest that DEGs and DAPs located on the substituted segments are direct factors causing the variation in seed weight in R217, including the genes related to the MAPK signaling pathway and the cell wall biosynthesis and modification pathway.

## 3. Discussion

Soybean is considered a dual functional crop of great economic value because of its high protein and oil content [53]. Seed weight is one of the traits that significantly affect soybean yield [3,9]. Establishing the genetic molecular mechanism underlying seed weight can offer strong clues for the breeding and genetic improvement of modern soybean varieties. In this study, a large seed CSSL R217 was selected from a CSSL population. The 100-seed weight, seed length, seed width, and seed weight per plant increased compared with its parent SN14. The substitution of ZYD06 chromosomal segments increased the seed size and weight of R217. The substituted segments of R217 were distributed on eight chromosomes, and 3203 genes were located on the substituted segments. These genes were mainly enriched in the MAPK signaling pathway, phenylpropanoid metabolic pathway, glycometabolism, and ubiquitination pathway. The transcriptomic analysis-based DEGs and proteomic analysis-based DAPs were also enriched in the MAPK signaling pathway and the cell wall biosynthesis and modification pathway at both S15 and S20 seed developmental stages, although the expression patterns of these DAPs and DEGs differed at the two stages. We thus uncovered two major potential pathways involved in seed weight control, including the MAPK signaling pathway and the cell wall biosynthesis and modification pathway.

### 3.1. CSSLs Facilitate Identifying Seed Size-Regulating Genes in Soybean

Because of its highly consistent genetic backgrounds, the CSSL is an efficient population for identifying the QTL related to seed weight, analyzing the interaction between the QTL, and improving varieties [54,55]. A few substituted CSSL segments can cause variations in seed size and weight. Identifying seed weight-related genes on the substituted segments directly contributes to reducing the workload compared to analyzing the whole soybean genome. CSSLs are widely used for studying multiple species, including rice, barley, tomato, and cotton [48,49,50,56]. A few CSSLs have also been developed in soybean [51,52,57]. In this study, a large seed CSSL R217 was selected from a CSSL population [51,52]. In total, 3203 genes were detected from the substituted segments of R217. The phenotypic variation of seed weight and seed size in R217 may result from the variation of these genes on the substituted segments at different levels, including functional mutations caused by changes in the genomic sequence, and differences in transcriptional-level expression and proteomic-level expression. Data obtained through RNA-seq and proteomic analyses revealed that all genes on the substituted segments were primarily enriched in glycometabolism, the MAPK signaling pathway, and the phenylpropanoid metabolic pathway. Based on the data of transcriptomic and proteomic analyses, several DEGs and DAPs were located on the substituted segments at S15 and S20. Finally, 23 DEGs and 2 DAPs were selected to participate in the MAPK signaling pathway and the cell wall biosynthesis and modification pathway and were considered key candidate genes related to seed size control (Figure 5 and Figure 6; Appendix A).

### 3.2. Integrated Transcriptomic and Proteomic Analyses Reveal the Seed Weight-Related Regulatory Pathways

Seed weight and size are complex quantitative traits that are controlled by many minor genes [1,2]. Therefore, cloning as many genes as possible that regulate seed weight, revealing the regulatory mechanism controlling seed weight, and establishing a seed weight regulatory network are challenging. Multi-omics analysis was recently used to identify and uncover essential components of stress responses, senescence, and yields in multiple economically crucial crops, including rice, maize, peanut, wheat, cotton, and soybean [58,59,60,61,62,63,64,65,66]. Thus, multi-omics analysis is among the effective methods for identifying seed weight-related genes and revealing their associated regulatory mechanisms. However, multi-omics analysis is currently only applied to study pollen development, seed storage profiles, and abiotic stress in soybean [18,66,67], and its application to seed weight in soybean has not been reported. In this study, transcriptome and proteomic analyses of R217 and SN14 were performed at S15 and S20 stages, respectively. In total, 2643 DEGs and 208 DAPs were detected at S15 and 1934 DEGs and 1248 DAPs were detected at S20. To more accurately and comprehensively reveal the seed weight-associated regulatory mechanism, we integrated the transcriptomic and proteomic data. Two seed weight-regulating pathways were uncovered, namely the MAPK signaling pathway and cell wall biosynthesis pathway. Moreover, 14 genes involved in the MAPK signaling pathway and 11 genes involved in the cell wall biosynthesis pathway were considered as seed weight-controlling candidate genes. These genes were all located on the substituted segments of R217. Genomic sequencing analysis will be performed based on the whole-genome re-sequencing data of CSSL [51,52], which will help further identify crucial seed weight-controlling genes from the aforementioned candidate genes.

### 3.3. Potential Regulatory Mechanisms Involved in Seed Size Control

The MAPK signaling pathway plays a crucial role in eukaryotes. This pathway has been identified to be involved in plant immunity, response to environmental stresses, and normal growth and development [21]. One of the crucial functions of the aforementioned pathway is controlling seed size [20,68]. Three tiers of protein kinases collectively compose a complete MAPK cascade [20,69]. The MAPK cascade OsMKKK10–OsMKK4–OsMPK6 module was revealed in rice [22,23,24], and the homologous module has been uncovered in *Arabidopsis* [26]. MAPK cascades are conserved among multiple species. The seed size-controlling pathways were further complemented based on the OsMKKK10-OsMKK4-OsMPK6 modules, including the ERECTA1-OsMKKK10-OsMKK4-OsMPK6-DST-cytokinin [70], ERECTA1-MAPK-DA1-UBP15 [22], OsMKK4-OsMAPK6-OsWRKY53 [30,31,71], MPK1-OsMKKK10-OsMKK4-OsMPK6 [25,27], and LecRK-VIII.2-OsMKKK10-OsMKK4-OsMPK6 modules [29]. Many genes were enriched in the MAPK signaling pathway in this study. In addition, several top DEGs and DAPs were also enriched in BR signaling and ubiquitin–proteasome pathways. Studies have reported that *OsWRKY53* in rice and its orthologous gene *WRKY33* in *Arabidopsis* are key genes regulating seed size that are involved in the BR signaling pathway [30,71]. The DA1–UBP15 module plays a crucial role in controlling seed size by participating in ubiquitin–proteasome pathways [72,73]. On combining the results of the previous studies and the present study, we noted that the candidate genes detected might control the seed size and weight by regulating the expression of MAPK-WRKY and MAPK–BR-related genes.

The main components of cell walls are pectin, cellulose, hemicellulose, and lignin, with lignin present between two cellulose molecules [74,75]. Cellulose is mainly composed of glucose, pectin is composed of a group of galacturonic acids, and lignin is mainly composed of phenylpropane. Cell walls are essential for vegetative development [32,76]. They provide structural support for plant cells, protect cell walls from abiotic and biotic stresses, and determine the final cell morphology [32,44]. *GW2* controlled the grain size of wheat by regulating the expression of cell wall synthesis-related genes and affected the synthesis of cell walls in wheat [36]. In barley, two pectin methylesterase genes *OVULE PECTIN MODIFIER 1* (*OPM1*) and *OPM2* controlled grain size by regulating homogalacturonan accumulation [42]. In *Arabidopsis*, the α-*XYLOSIDASE1* (*XYL1*) gene encodes the only α-xylosidase acting on xyloglucans, which are the major hemicellulose polysaccharides in the primary cell walls of dicots and non-graminaceous monocots. *XYL1* was regulated by the MADS box transcription factor *SEEDSTICK* to affect cell wall thickness and ultimately determine seed size [39]. *GS3.1* regulates the grain size of rice by controlling metabolic-flux distribution between flavonoid and lignin metabolons [77]. In summary, the cell wall composition and modification regulate seed size. However, details of the regulatory mechanisms underlying these cell wall-related genes have not been reported. Several top DEGs and DAPs in this study were enriched in cell wall modification, the cell wall polysaccharide metabolic process, the pectin metabolic process, and phenylpropanoid biosynthesis. These results suggested that genes encoding cell wall biological process-related top DEGs and DAPs are crucial seed size-controlling candidate genes. The mechanism regulating seed weight through cell wall regulation will be further explored in future work.

Moreover, MAPK cascades are involved in cell wall integrity regulation [32]. The MAP-kinase 3 phosphorylates UDP–glucose dehydrogenase and provides UDP–sugar for cell wall biosynthesis in *Arabidopsis* [78]. MPK4 affects cellulose content by negatively regulating L-arabinose synthesis for cell wall formation in *Arabidopsis* [79]. Based on these reported works, we inferred that MAPK cascades regulate cell wall biosynthesis, but the cooperative regulation of these cascades and the cell wall process controlling seed size remain unclear. Therefore, based on the results of our integration of transcriptomic and proteomic analyses, we believed that the MAPK cascades and cell wall biosynthesis and modification may co-regulate seed weight and size (Figure 6; Appendix A).

## 4. Materials and Methods

### 4.1. Plant Materials

A large seed CSSL R217 was selected from the soybean CSSL population. The donor parent of the CSSL population was wild soybean ZYD00006, and the recurrent parent was the cultivated soybean SN14 [51,52]. These are varieties from Heilongjiang Province, China. The CSSLs have been re-sequenced [52], and were grown at Xiangyang experimental Farm of Northeast Agricultural University in Harbin, China (45.75° N, 126.53° E). Each row was 5 m in length and the spacing distance was between plants was 6 cm and that between the rows was 0.6 m. After the seeds naturally matured, the harvest was completed. The materials used for sequencing were planted in a greenhouse under the same conditions, and each material was replanted three times.

### 4.2. Seed Weight and Size Evaluation

Three individuals with dry seeds were randomly selected from R217 and SN14 after the soybean plants matured. The 100-seed weight, seed length, seed width, and seed weight per plant of R217 and SN14 were estimated from 2018 to 2022. The 100-seed weight was measured by randomly selecting 100 high-quality seeds for each individual. The seed length and width were estimated using an average of 5 seeds. The seed weight per plant was evaluated by weighing all the seeds produced by a single plant.

### 4.3. RNA-Seq Analysis

Total RNA was extracted from three independent biological samples of R217 and SN14 at S15 and S20. Total RNA was isolated using the TRIzol reagent (Invitrogen, Carlsbad, CA, USA) according to the manufacturer’s protocol. mRNA was purified and fragmented into small pieces from 50 μg of total RNA. mRNA was reverse transcribed to produce cDNA for RNA-seq research. Applied Protein Technology (APTBIO, Shanghai, China) performed RNA-seq analysis by using an Illumina NovaSeq 6000 system to produce 150-bp pair-end (PE150) reads. fastq was used to perform basic statistics on raw seed quality. The clean reads were mapped to the soybean genome by using Hisat2 (version 2.2.1) [80]. To measure the gene expression levels, we calculated the Fragments Per Kilobase of the transcript sequence per Millions base pairs sequenced values [81]. DESeq2 was further used to analyze the gene expression profile and detect differentially expressed genes (DEGs) [82]. The criteria were ¦log2(fold change)¦ > 1.0 and corrected *p* value (padj) < 0.05.

### 4.4. Protein Extraction and Digestion

Lysis and protein extraction in R217 and SN14 were performed using the lysis buffer SDT (4% SDS, 100 mM Tris-HCl, 1 mM DTT, pH 7.6). The BCA Protein Assay Kit (Bio-Rad, Hercules, CA, USA) was used to quantify the protein amount. The filter-aided sample preparation procedure described by Matthias Mann was used for performing trypsin-mediated protein digestion. The digested peptides of each sample were desalted on C18 cartridges (Empore™ SPE Cartridges C18 (standard density), volume 3 mL, bed I.D. 7 mm, Sigma, St. Louis, MO, USA), concentrated through vacuum centrifugation, and reconstituted in 40 µL of 0.1% (*v*/*v*) formic acid. The mixture of 20 µg of each protein sample and 5× loading buffer was boiled for 5 min. The proteins were separated using a 12.5% SDS-PAGE gel at a constant current of 14 mA for 90 min. Coomassie Blue R-250 staining was performed to visualize the protein bands.

### 4.5. LC-MS/MS Analysis

The LC-MS/MS analysis was performed using the timsTOF Pro mass spectrometer (Bruker, Billerica, MA, USA), which was coupled with Nanoelute (Bruker Daltonics) for 60/120/240 min. A reverse phase trap column (Thermo Scientific, Waltham, MA, USA, Acclaim PepMap100, nanoViper C18, 100 μm × 2 cm) was applied to load the peptides, and the reverse phase trap column was connected to the C18-reversed phase analytical column (Thermo Scientific Easy Column, inner diameter, 75 μm; length, 10 cm; resin, 3 μm) in 0.1% formic acid. The IntelliFlow technology was used to separate a linear gradient of the mixture of 0.1% formic acid and 84% acetonitrile at a flow rate of 300 nL/min. The mass spectrometer was used with the positive ion mode, and ion mobility MS spectra were collected over a mass range of *m*/*z* 100–1700 and 1/k0 of 0.6–1.6. The mass spectrometer operated 10 cycles of PASEF MS/MS with a target intensity of 1.5 k, and the threshold was set as 2500. Release occurred for 40 min under active exclusion.

### 4.6. Proteomic Data Processing

The MS raw data for the R217 and SN14 samples were searched and combined using MaxQuant 1.5.3.17 software for the identification and quantitation analyses [83]. A maximum of two missed cleavages were allowed for tryptic peptides. The modification pattern, including carbamidomethyl (C) and oxidation (M), was added, and the MS/MS tolerance was set to 20 ppm. Proteins were annotated by searching the Uniprot_Glycine database (downloaded from https://www.uniprot.org/taxonomy/3847, with 85,057 protein sequences, accessed on 4 September 2021), and the database pattern was set to Target-Reverse. The protein FDR ≤ 0.01 was required for positive protein hits. The intensity-based absolute quantification method and the label-free quantification method in MaxQuant 1.5.3.17 software were used for protein quantification [83]. The criterion of FC > 2 or <0.5 (*p* < 0.05) was established to identify differentially accumulated proteins (DAPs) in Perseus [84] and R (https://www.r-proje ct.org/, accessed on 22 October 2022).

### 4.7. Bioinformatics Analysis

Principal component analysis was performed to evaluate variations in DEG expression by using the stats package in R. The information on GO annotation was obtained from the database of SoyBase (https://soybase.org, accessed on 28 October 2022). The annotated information on DAPs was based on the UniProtKB/Swiss-Prot (https://ww.uniprot.org/, accessed on 28 October 2022). The results of enrichment analyses of both DEGs and DAPs were from Gene Ontology (GO; http://geneontology.org/, accessed on 8 November 2022) and Kyoto Encyclopedia of Genes and Genomes (KEGG; https://www.kegg.jp/, accessed on 8 November 2022) databases. All quantified proteins were considered the background dataset, and the enrichment analyses were performed using Fisher’ exact test. To further adjust the derived *p* values, the Benjamini–Hochberg correction for multiple testing was performed. The GO and KEGG enrichment analyses were performed using the Clusterprofiler (version 4.0.5) [85].

### 4.8. Statistical Analysis

Significance analyses were performed using one-way ANOVA in SPSS Statistics17.0 software (International Business Machines Corporation, New York, NY, USA), and the threshold was set to *p* < 0.05. The box plots, volcano plots, and histograms were drawn using the “ggplot2” [86] and “ggpubr” (http://mysql.orst.edu/pub/cran/web/packages/ggpubr/index.html, accessed on 20 July 2023) packages in R version 4.0.2 (https://www.r-project.org/, accessed on 20 July 2023). The heatmaps were plotted using the “*pheatmap* package” (https://cran.r-project.org/web/packages/pheatmap/index.html, accessed on 26 July 2023) in R version 4.0.2 (https://www.r-project.org, accessed on 26 July 2023).

## 5. Conclusions

This study reveals the potential seed weight regulatory pathways in soybean, including the MAPK signaling pathway and cell wall biological processes (Figure 6; Appendix A). CSSL R217 was selected from the CSSL population. The 100-seed weight, seed length, seed width, and seed weight per plant of R217 increased compared to SN14. Integrated analysis of the transcriptomic and proteomic results indicated that these DEGs and DAPs were primarily enriched in the MAPK signaling pathway and cell wall biosynthesis pathway, and 59 candidate genes associated with 100-seed weight were predicted. Moreover, the gene analysis on the instituted segments of R217 revealed 25 candidate genes associated with 100-seed weight located on these segments. In conclusion, a potential seed weight regulatory module was uncovered in this work, and 25 candidate genes were predicted to be involved in the two seed weight control pathways (Figure 5; Appendix A). These results provide new insights into the molecular mechanisms of soybean seed weight.

## Figures and Tables

**Figure 1 plants-13-00908-f001:**
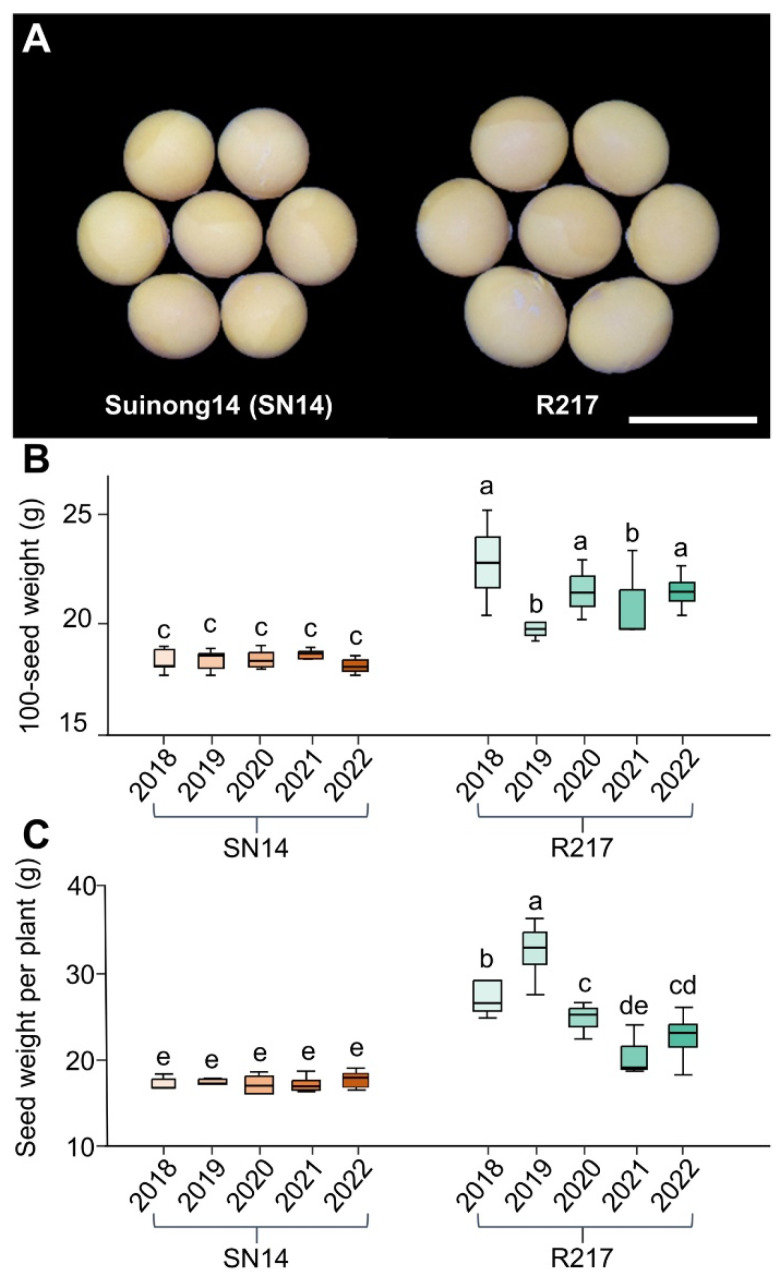
The comparison of 100-seed weight and seed weight per plant between SN14 and large-seed CSSL R217 for 5 years. (**A**) The seed sizes of SN14 and R217. Bar = 1 cm. (**B**,**C**) The phenotypic data of 100-seed weight and seed weight per plant for SN14 and R217 from 2018 to 2022. In (**B**,**C**), different lowercase letters (a, b, c, d, and e) indicate significant differences, the threshold is set to *p* < 0.05, and identical letters suggest insignificant differences in one-way ANOVA.

**Figure 3 plants-13-00908-f003:**
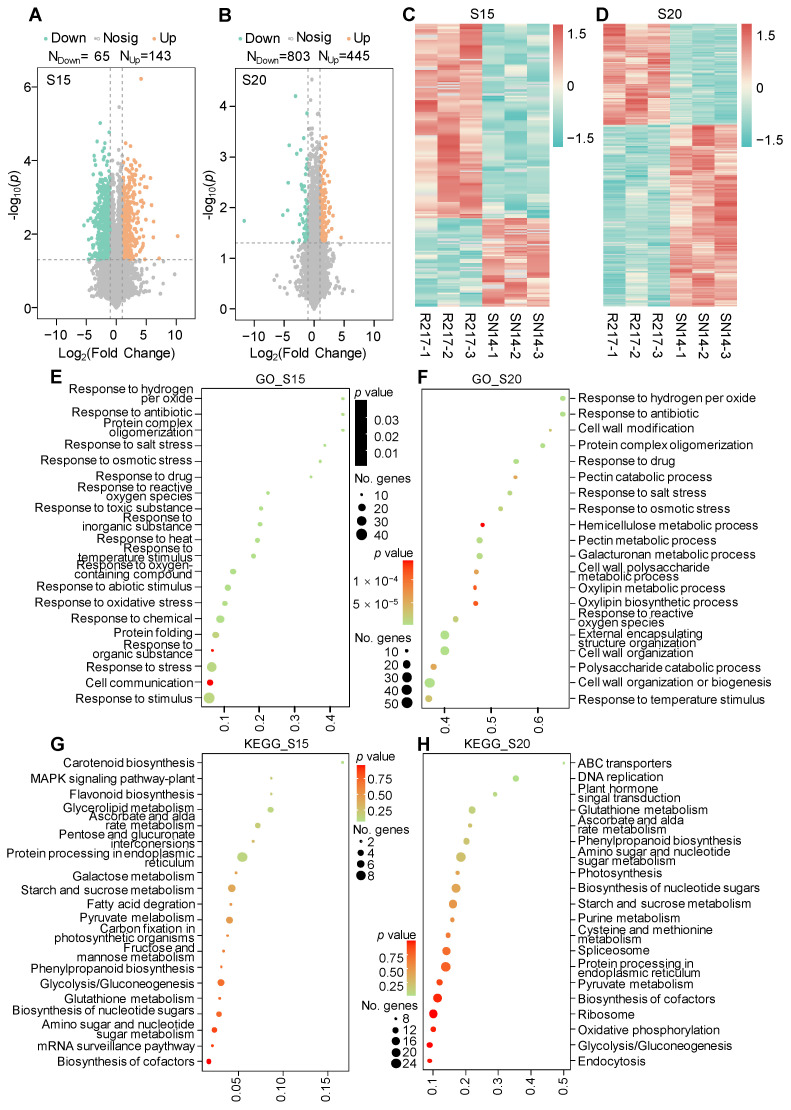
The DAP analysis, GO enrichment analysis, and KEGG enrichment analysis of DAPs in R217 and SN14 at S15 and S20 seed developmental stages. (**A**,**B**) Volcano maps of all DAPs at the S15 and S20 seed developmental stages. Down—DEGs with downregulated expression. Up—DEGs with upregulated expression. Nosig—genes without differential expression in RNA-seq analysis. N_Down_—number of downregulated DAPs. N_Up_—number of upregulated DAPs. (**C**,**D**) The heatmap of all DAPs in R217 and SN14 at S15 and S20. (**D**,**E**) The GO enrichment analysis of the top DAPs at S15 and S20. (**F**,**G**) The KEGG enrichment analysis of the top DAPs at S15 and S20. In (**E**–**H**), the vertical coordinates represent the GO terms enriched with genes from the substituted segments. The horizontal axis represents the gene ratio.

**Figure 4 plants-13-00908-f004:**
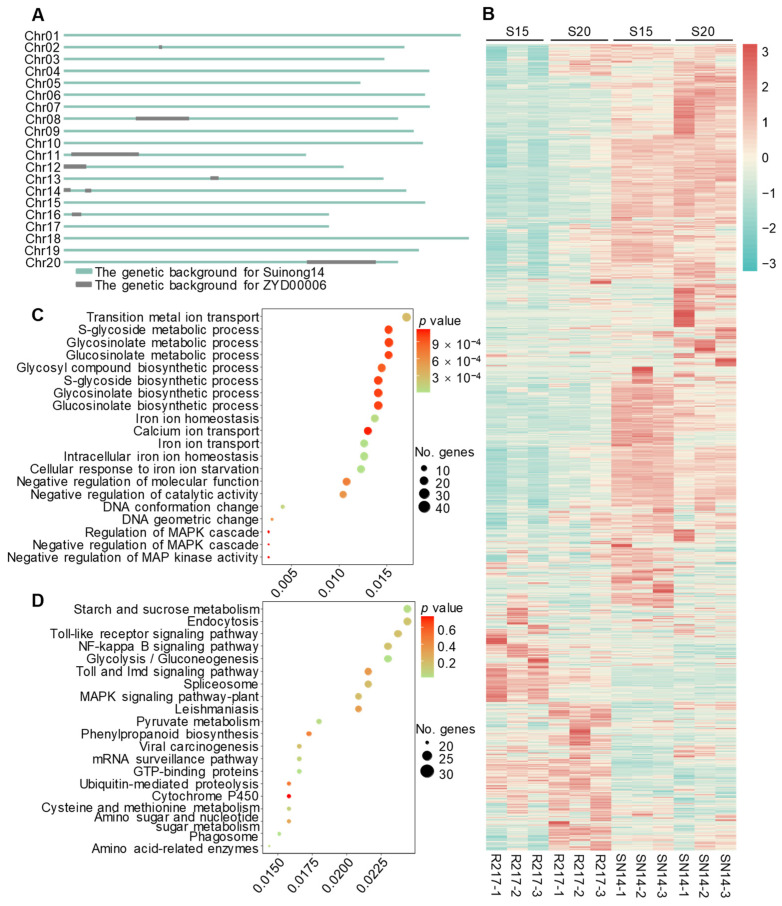
Gene analysis and enrichment analyses on the substituted segments of R217. (**A**) Genome diagram of R217. The distribution of ZYD00006 substituted segments on the R217 genome. The green lines represent the SN14 genome and the gray boxes represent the ZYD00006 genome. (**B**) The heatmap of all genes on the substituted region of R217 at S15 and S20 seed developmental stages. S15 and S20 represent the 15th and 20th days after fertilization, respectively. (**C**) The GO enrichment analysis of genes on the substituted segments of R217. (**D**) The KEGG enrichment analysis of all genes on the substituted region of R217. In (**C**,**D**), the vertical coordinates represent the GO terms enriched with genes from the substituted segments. The horizontal axis represents the gene ratio. No. genes—the number of genes.

**Figure 5 plants-13-00908-f005:**
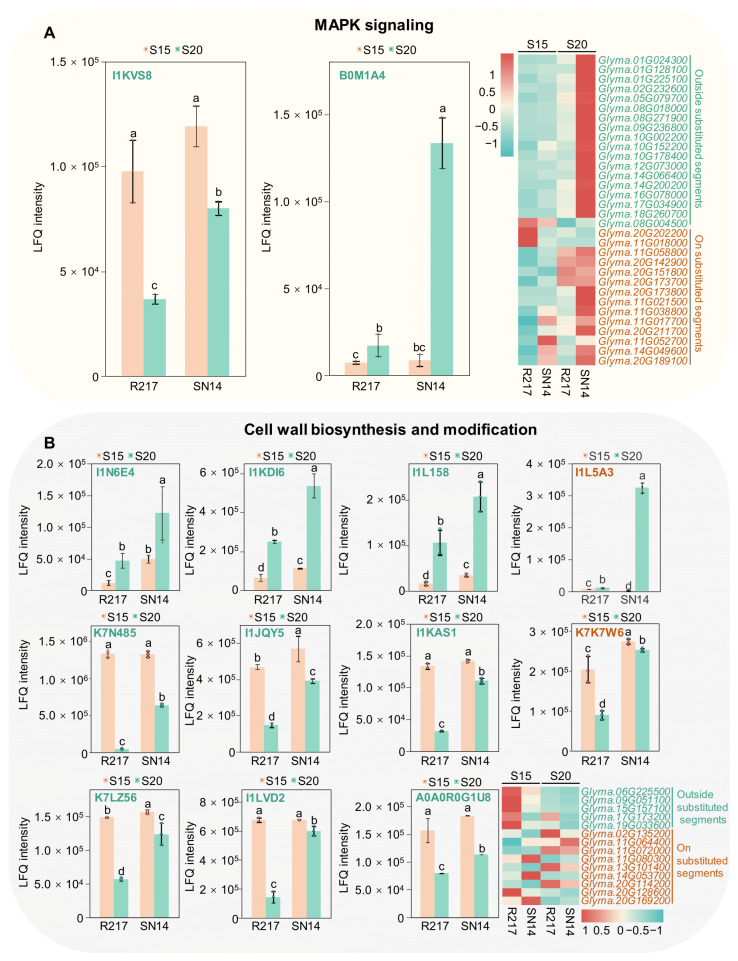
The expressed pattern of DAPs and DEGs involved in the 100-seed weight-related pathway at S15 and S20 seed developmental stages. (**A**,**B**) DAPs and DEGs involved in the MAPK signaling pathway at S15 and S20 seed developmental stages. The green font represents the genes or protein on the substituted segments. The tangerine font represents the genes or protein outside the substituted segments. Different lowercase letters (a, b, c, and d) indicate significant differences and identical letters suggest insignificant differences in one-way ANOVA.

**Figure 6 plants-13-00908-f006:**
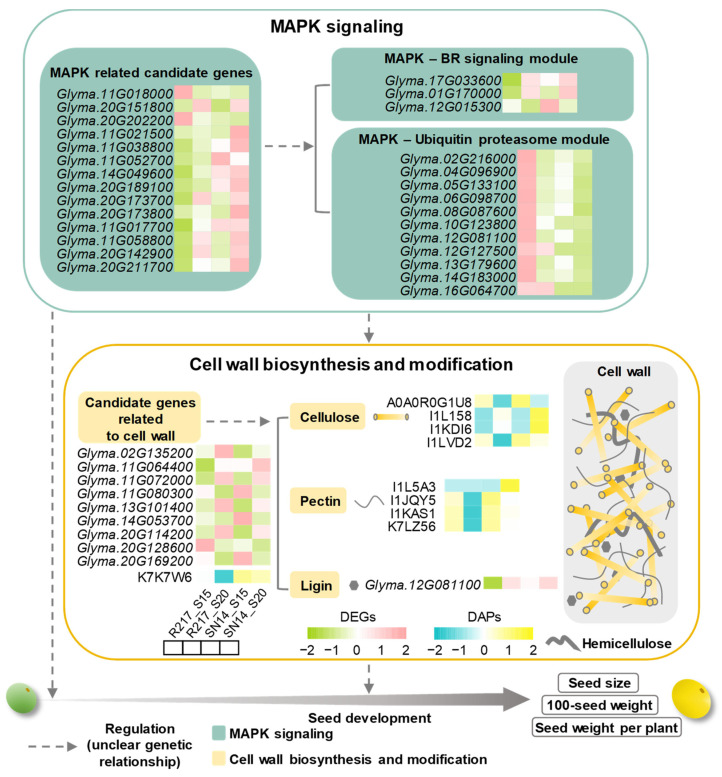
Potential seed weight regulatory mechanisms in soybeans. The MAPK signaling and the cell wall biosynthesis and modification pathways are potential seed weight-controlling pathways in soybean. The candidate genes were selected based on the data of the integrated transcriptomic and proteomic analyses in SN14 and R217 at S15 and S20 seed developmental stages. The dashed lines represent an unclear genetic relationship.

## Data Availability

All data are available in the manuscript or the Appendix A. The RNA-seq data have been deposited in the NCBI at the Sequence Read Archive (SRA) database under the accession numbers of PRJNA1031595. The mass spectrometry proteomic data have been deposited to the ProteomeXchange Consortium (http://proteomecentral.proteomexchange.org, the announcement date is 25 October 2023.) through the iProX partner repository with the data set identifier PXD046403. All plant materials reported in this study are available upon completion of the material transfer agreement.

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
