# Peer review of "Integrated Transcriptomic and Proteomic Characterization of a Chromosome Segment Substitution Line Reveals the Regulatory Mechanism Controlling the Seed Weight in Soybean"

_plants, 2024, doi:10.3390/plants13060908_

Round 1

Reviewer 1 Report (Previous Reviewer 2)

Comments and Suggestions for Authors

I see no significant improvement in the revised version of manuscript plants-2891224, making it unsuitable for publication in the Plants.

Comments on the Quality of English Language

Extensive editing of English language required.

Author Response

Dear Editors and Reviewers:

Thank you for the referee’s comments concerning our manuscript entitled ‘Integrated transcriptomic and proteomic characterization of a chromosome segment substitution line reveals the regulatory mechanism controlling the seed weight in soybean’. We have carefully revised the manuscript in accordance with the reviewer's comments and have also conducted language editing on the revised manuscript. We look forward to this revised manuscript being accepted by the editor and reviewers. Thank you and best regards.

Yours sincerely,

Jianan Zou and Jinhui Wang

Answer to reviewer 1:

Qingshan Chen and Jinhui Wang. I see no significant improvement in the revised version of manuscript plants-2891224, making it unsuitable for publication in the Plants.

Answer: We sincerely appreciate this reviewer’s comments and suggestions. Despite the considerable efforts invested in our work, we regret to acknowledge the absence of this reviewer’s endorsement for our work.

Reviewer 2 Report (Previous Reviewer 1)

Comments and Suggestions for Authors

The manuscript entitled " Integrated transcriptomic and proteomic characterization of a chromosome segment substitution line reveals the regulatory mechanism controlling the seed weight in soybean," has been substantially improved so that a reader can follow.

Authors have performed a transcriptomics and proteomics analysis on the chromosome segment substitution line (CSSL) R217 and its recurrent parent Suinong14 (SN14). The approach is proper and interesting for scientist in the field.

However, important issues still exist that make it not appropriate for publishinhg in its current state.

My main concerns have to do with the non-availability of the data. Such studies should provide all their results so as that the reader can reproduce some of the results. Thus, I think that all lists with all DEGs and DAPs, for each strain, for each time point should be given in the form of supplementary tables or material in xl files (perhaps along with values and criteria of selection).  

 Chapter 2.4 should be elaborated in order to give more details on the procedure (or create a new methods section)

Lines 240 and 254 : Please give location and the reason for the selection

Lines 312 313 : Please give the basis on which this selection was made

There are many grammatical and syntax errors throughout the whole manuscript (eg lines 26, 82, 105, 122, 123, 126, 143).  The manuscript needs editing by a n English native speaker.

Minor points

The manuscript looks slopy since “leftovers’ from the previous version do exist (line 114, lines 156 157 , bracket and full stop  in line 159)

Line 17: “revealed” is an overstatement. Better use “investigated”

Line 129, 133: The use of etc is not proper here. What does it mean? What is the reader expected to think ? Why authors do not give more or less pathways. A threshold criterion should be given.

Line 178 : Unacceptable syntax

Comments on the Quality of English Language

There are many grammatical and syntax errors throughout the whole manuscript (eg lines 26, 82, 105, 122, 123, 126, 143).  The manuscript needs editing by a n English native speaker.

Line 178 : Unacceptable syntax

Author Response

Thank you for the reviewer’s comments interesting our manuscript entitled ‘Integrated transcriptomic and proteomic characterization of a chromosome segment substitution line reveals the regulatory mechanism controlling the seed weight in soybean’. We have carefully revised the manuscript in accordance with the reviewer's comments and have also conducted language editing on the revised manuscript. We look forward to this revised manuscript being accepted by the editor and reviewers. Thank you and best regards.

Yours sincerely,

Qingshan Chen, Jianan Zou and Jinhui Wang

Answer to reviewer2:

The manuscript entitled " Integrated transcriptomic and proteomic characterization of a chromosome segment substitution line reveals the regulatory mechanism controlling the seed weight in soybean," has been substantially improved so that a reader can follow.

Answer: We are pleased to have received the reviewer’s approval of our improvements and positive comments on our work. Building on this feedback, we have further enhanced our manuscript in line with the editor’s and peer reviewer’s suggestions and recommendations.

Authors have performed a transcriptomic and proteomics analysis on the chromosome segment substitution line (CSSL) R217 and its recurrent parent Suinong14 (SN14). The approach is proper and interesting for scientist in the field.

Answer: We fully agree with the Reviewer that the chromosome segment substitution line (CSSL) is a material that is suitable for studying the mechanism of complex trait variation, as each CSSL genome carries only few substituted segments. The R217 in this work carries a small amount of substituted segments. Additionally, the seed size of R217 is significantly higher than that of SN14, making R217 a very suitable material for studying the mechanism of seed size variation. Furthermore, multi-omics analysis is an effective approach to explain seed size variation from multiple perspectives. This work is based on an integrated transcriptomic and proteomic comparative analysis of R217 and SN14 to reveal the molecular mechanism of soybean seed size. Finally, All the formatting errors and grammar errors in the manuscript have been corrected and the manuscript has been re-edited for language.

However, important issues still exist that make it not appropriate for publishing in its current state.

Answer: We are very sorry for some errors that were left over from last time. We have once again carefully corrected the errors in the manuscript and figure legends. However, please allow me to explain that apart from the grammar errors, some formatting issues were actually caused by the version of the PDF you used for reviewing, which was converted from our tracked changes version, leading to some formatting errors during the conversion. At this submission, we will carefully check whether the version of the manuscript is correct to avoid any formatting conversion errors.

My main concerns have to do with the non-availability of the data. Such studies should provide all their results so as that the reader can reproduce some of the results. Thus, I think that all lists with all DEGs and DAPs, for each strain, for each time point should be given in the form of supplementary tables or material in xl files (perhaps along with values and criteria of selection).  

Answer: I fully agree with the reviewer’s suggestion. To demonstrate the availability of the data in this work, we have added three supplementary tables, Table S1, Table S2, and Table S3. Table S1 lists the position information of the R217 substituted segments. Table S2 provides the expression and protein abundance data of 59 candidate genes for 100-seed weight in Figure 5. Table S3 presents the expression and protein abundance information of genes that may be involved in the regulatory pathway of 100-seed weight in Figure 6.

Chapter 2.4 should be elaborated in order to give more details on the procedure (or create a new method section)

 Answer: I greatly appreciate and agree with the suggestions made by the reviewer. We have provided a more detailed description of the results in Chapter 2.4, where the modified content is indicated in red font. Moreover, the location information of the substituted segments of CSSL R217 has been included in Table S1.

Lines 240 and 254 : Please give location and the reason for the selection

Answer: I apologize for the confusion caused to the reviewer by the content in these two places. We have provided clearer descriptions for “On a location other than the 240 substituted segments, 2 DAPs and 18 DEGs that participated in the MAPK signaling 241 pathway were selected (Figure 5A)” and “Furthermore, 9 DAPs and 5 DEGs involved in cell wall biosynthesis 254 and modification were selected outside the substituted segments of R217 (Figure 5B)”, which have been respectively modified to “Furthermore, the analysis results of the DEGs and DAP expression patterns based on FPKM and LFQ intensity data indicate that 2 DAPs and 18 DEGs are involved in the MAPK signaling pathway in regions outside of the substituted segments (Figure 5A)” and “Furthermore, the analysis results of gene expression patterns based on FPKM and LFQ intensity data indicate that 9 DAPs and 5 DEGs outside the substituted segments were involved in cell wall biosynthesis and modification (Figure 5B)”. In addition, we explained that the reason for selecting these candidate genes is based on the expression pattern analysis of FPKM data and LFQ intensity data.

Lines 312 313: Please give the basis on which this selection was made

Answer: I greatly appreciate and agree with your opinion, and we have also made additions and modifications in this manuscript. The sentence located in Lines 312 and 313 was “Finally, 23 DEGs and 2 DAPs to participate in the MAPK signaling pathway and the cell wall biosynthesis and modification pathway were selected, and these DEGs and DAPs were considered key candidate genes related to seed size control”. This sentence has been modified to “Finally, 23 DEGs and 2 DAPs located on the substituted segments participate in the MAPK signaling pathway and cell wall biosynthesis and modification pathways were selected based on analysis of expression patterns, and these DEGs and DAPs were considered key candidate genes related to seed size control (Figures 5 and 6; Tables S2 and S3)” in the latest version of this manuscript. In summary, we have provided two reasons for selecting these candidate genes. Firstly, the candidate genes were identified based on the expression pattern analysis of FPKM data and LFQ intensity data. Additionally, these candidate genes are located on the substituted segments of R217. Finally, we have supplemented the expression data and protein abundance data of candidate genes in supplementary Tables.

There are many grammatical and syntax errors throughout the whole manuscript (eg lines 26, 82, 105, 122, 123, 126, 143).  The manuscript needs editing by an English native speaker.

Answer: We are very sorry for the grammatical and syntax errors in the manuscript. Once again, we have enlisted professional language editors to re-edit the manuscript, including the grammatical and syntax errors in lines 26, 82, 105, 122, 123, 126, and 143.

Minor points

The manuscript looks sloppy since “leftovers’ from the previous version do exist (line 114, lines 156 157, bracket and full stop in line 159)

Answer: We feel ashamed for the errors left from last time. We have corrected the errors in the content of manuscript and figure legends. Indeed, the errors in “line 114, lines 156 157, bracket and full stop in line 159” are caused by the version of the PDF you used for reviewing, which was converted from our tracked changes version, leading to some formatting errors during the conversion. We will carefully check the version before this submission.

Line 17: “revealed” is an overstatement. Better use “investigated”

Answer: I agree with the reviewer's suggestion, and I have modified "revealed" to "investigated" in Abstract.

Line 129, 133: The use of etc is not proper here. What does it mean? What is the reader expected to think? Why authors do not give more or less pathways. A threshold criterion should be given.

Answer: Thank you for the reviewer's comments. We also agree that the use of "etc" was inappropriate and have deleted it in the manuscript. What we want to convey is the pathways mainly enriched based on GEGs and DAPs, not all pathways, so it is sufficient to list the more significant pathways without using "etc".

Line 178: Unacceptable syntax

Answer: We deeply apologize for the grammar issue in the sentence on line 178, which may cause difficulty for readers to comprehend. We have made modifications to this sentence as “At S15, response to stress-related GO terms are mainly enriched (Figure 3E). However, at S20, cell wall biosynthesis and modification related GO terms were mainly enriched, including pectin catabolic or metabolic process, cell wall modification, and cell wall organization or biogenesis at S20, as determined based on the GO enrichment analysis of DAPs (Figure 3F)”.

Reviewer 3 Report (New Reviewer)

Comments and Suggestions for Authors

Dear Authors,

The manuscript: "Integrated transcriptomic and proteomic characterization of a chromosome segment substitution line reveals the regulatory mechanism controlling the seed weight in soybean" contains valuable research on the molecular characterization of mechanisms regulating soybean seed weight. The manuscript was written very carefully. Below are some minor comments:

- The abstract contains all the most important information relating to the experiment

- Introduction - written reliably and in detail, it provides a good introduction to the topic of the publication. In the first paragraph (L: 34), it is advisable to add information about the current area of soybean cultivation and harvest, as well as a brief description.

- Results - are well described, the figures are of good quality and present the data in an interesting way

- Discussion: Refers in detail to the obtained research results, In L: 280 there is no citation of the literature

- Materials and Methods - 4.1. Plant Material - Information about soil and climate conditions is missing, please provide it.

- Conclusions - There is no information at the end of this chapter about the application of the obtained research results

Author Response

Thank you for the reviewer’s comments interesting our manuscript entitled ‘Integrated transcriptomic and proteomic characterization of a chromosome segment substitution line reveals the regulatory mechanism controlling the seed weight in soybean’. We have carefully revised the manuscript in accordance with the reviewer's comments and have also conducted language editing on the revised manuscript. We look forward to this revised manuscript being accepted by the editor and reviewers. Thank you and best regards.

Yours sincerely,

Qingshan Chen, Jianan Zou and Jinhui Wang

The manuscript: "Integrated transcriptomic and proteomic characterization of a chromosome segment substitution line reveals the regulatory mechanism controlling the seed weight in soybean" contains valuable research on the molecular characterization of mechanisms regulating soybean seed weight. The manuscript was written very carefully. Below are some minor comments:

Answer: I appreciate the reviewer's affirmation of this work.

- The abstract contains all the most important information relating to the experiment

Answer: Thank the reviewer for the high evaluation of the abstract.

- Introduction - written reliably and in detail, it provides a good introduction to the topic of the publication. In the first paragraph (L: 34), it is advisable to add information about the current area of soybean cultivation and harvest, as well as a brief description.

Answer: We appreciate the reviewer’s evaluation and comments. We agree and have added information on the current area of soybean cultivation and harvest at the corresponding position in line 34, which is cited from the relevant information source and highlighted in red font in the manuscript.

- Results - are well described, the figures are of good quality and present the data in an interesting way

Answer: I am very grateful to the reviewer for the comments on the results section.

- Discussion: Refers in detail to the obtained research results, In L: 280 there is no citation of the literature

Answer: I agree with the reviewer's suggestion. We have made appropriate modifications to this sentence to make the description more accurate, and have provided the citation of the literature in the corresponding section of the manuscript (Line 280).

- Materials and Methods - 4.1. Plant Material - Information about soil and climate conditions is missing, please provide it.

Answer: Thank the reviewer for the suggestion. I agree with the reviewer's comments, and the detailed planting conditions have been added to the part of 4.1. Plant Material. The added information has been highlighted in red font.

- Conclusions - There is no information at the end of this chapter about the application of the obtained research results

Answer: I fully agree with the reviewer's comments. We have summarized and condensed this work, providing key information in the conclusion. In addition, the end of this chapter about the application of the obtained research results has been also supplemented.

This manuscript is a resubmission of an earlier submission. The following is a list of the peer review reports and author responses from that submission.

Round 1

Reviewer 1 Report

Comments and Suggestions for Authors

I will be glad to consider a revised version of the manuscript provided that figures, figure legends, and references to the figures within manuscript, are all put in place and in an order. As it is now the manuscript, it is not readable. Figure 2 is twice depicted in the mauscript with different figure legends. There is missing information for fig 4. Please add title in the X and Y axes where absent. 

Comments on the Quality of English Language

no comments

Reviewer 2 Report

Comments and Suggestions for Authors

In this manuscript entitled "Integrated Transcriptomic and Proteomic Characterization of a Chromosome Segment Substitution Line Reveals the Regulatory Pathways Controlling Seed Weight in Soybean," the authors report the results of transcriptomic and proteomic analyses conducted on the chromosome segment substitution line (CSSL) R217 and its recurrent parent Suinong14 (SN14). While the topic is of interesting, the manuscript suffers from several serious problems, making it unsuitable for publication in this esteemed journal.

1.     The manuscript lacks conclusiveness, providing limited information for the readers of the Plants, or even potentially offering a misleading interpretation. Although CSSL R217 exhibited higher 100 seed-weight than its original parent SN14, there are at least 8 chromosome segments that differ between the two lines. It is necessary to include additional lines harboring the wild soybean chromosome segment, know contributes to the big seed size, in the experiment.

2.     There are numerous of mistakes in manuscript, significantly impacting its scientific quality.

3.     Page 4: As far as I understand, the results described in this page appears to be based on gene annotation for the substituted chromosomal segments of R217. However, I could not understand why authors stated, that ‘The expression patterns of all genes were analyzed based on the RNA-seq data and proteomic analysis data (Figure 2),’ and why Figure 2 was cited, which pertains to "The differentially expressed genes (DEGs) analysis and the DEGs GO and KEGG enrichment analyses on substituted segments of the R217 genome. "

4.     It has come to my attention that Figure 2 and Figure 3 are identical. I recommend that the authors rectify this redundancy and maintain a more rigorous attitude in preparing the scientific manuscript.

Comments on the Quality of English Language

See Comments and Suggestions for Authors, please.